# Habitat Conditions of the Microbiota in Ballast Water of Ships Entering the Oder Estuary

**DOI:** 10.3390/ijerph192315598

**Published:** 2022-11-24

**Authors:** Kinga Zatoń-Sieczka, Elżbieta Bogusławska-Wąs, Przemysław Czerniejewski, Adam Brysiewicz, Adam Tański

**Affiliations:** 1Department of Commodity, Quality Assessment, Process Engineering and Human Nutrition, Faculty of Food Sciences and Fisheries, West Pomeranian University of Technology in Szczecin, Ul. Kazimierza Królewicza 4 St., 71-550 Szczecin, Poland; 2Department of Microbiology and Applied Biotechnology, Faculty of Food Sciences and Fisheries, West Pomeranian University of Technology in Szczecin, Ul. Papieża Pawła VI St., 71-459 Szczecin, Poland; 3Institute of Technology and Life Sciences–National Research Institute, Falenty, 3 Hrabska Avenue, 05-090 Raszyn, Poland; 4Department of Hydrobiology, Ichthyology and Reproductive Biotechnology, Faculty of Food Sciences and Fisheries, West Pomeranian University of Technology in Szczecin, Ul. Kazimierza Królewicza 4 St., 71-550 Szczecin, Poland

**Keywords:** ballast water management, water environment, microorganisms, water transport, water chemistry, wastewater

## Abstract

Ballast water is a vector for the transfer of microorganisms between ecospheres that can subsequently have a negative impact on native species of aquatic fauna. In this study, we determined the microbiota and selected physicochemical properties of ballast water from long- and short-range ships entering a southern Baltic port within a large estuary in autumn and winter (Police, Poland). Microbiological tests of the ballast water samples were carried out according to ISO 6887-1, and physicochemical tests were performed according to standard methods. Low amounts of oxygen (1.6–3.10 mg/dm3 in autumn and 0.60–2.10 mg/dm3 in winter) were recorded in all ship ballast water samples, with pH (above 7.90) and PSU (above 1.20) were higher than in the port waters. Yeast, mold, *Pseudomonas* bacteria (including *Pseudomonas fluorescens*), and halophilic bacteria as well as lipolytic, amylolytic, and proteolytic bacteria were found in the ballast water samples. Heterotrophic bacteria and mold fungi (log. 2.45–3.26) dominated in the autumn period, while *Pseudomonas* bacteria (log. 3.32–4.40) dominated in the winter period. In addition, the ballast water samples taken during the autumn period were characterized by a statistically significantly higher (*p* < 0.1) abundance of microorganisms (log 1.97–2.55) than in the winter period (log 1.39–2.27).

## 1. Introduction

Marine transport is considered a major threat to aquatic ecosystems worldwide. This is mainly associated with the discharge of ballast waters in ports to maintain the stability of the vessel and the threat of inadvertent transfer of alien species [1,2]. The process of ballast water exchange (along with the microorganisms, animals, and plants therein) occurs in coastal regions and can affect local aquatic industries, for example, aquaculture. It is estimated that exchanged ballast water is a vector for as much as a third of alien organisms [3,4]. Some of them are invasive [5] and pose a threat to native organisms and their habitats [6].

The seemingly harmless appearance of non-native species in new environments can become a threat as they establish themselves and successfully take over an ecosystem at least partially similar to their native one. Such species, when unconstrained by natural enemies, slowly expand their range and begin to take on an invasive character as they compete for habitat and food. Eventually, this leads to the displacement of the native species, which is not only a problem for individual species but also for the biodiversity of entire regions in the world [7,8,9,10]. This is particularly dangerous for enclosed and semienclosed water bodies, which have a naturally low biodiversity and at the same time are heavily used for marine transport [11]. One such marine reservoir is the Baltic Sea [11,12]. The threat of the emergence of new species is also compounded by things seemingly not related to geography, namely, microorganisms with an unknown impact on the local environment. Among other things, in order to control the appearance of microorganisms in ballast water, the International Convention on the Control and Management of Ship’s Ballast Water and Sediment (BWM) was established [13]. This document imposed an obligation on shipowners to conduct ballast water tests for the presence of the indicator bacteria *E. coli* and *Enterococcus* and the pathogenic bacteria *Vibrio cholerae*. This aroused great interest among scientists, which resulted in many works focused on ballast microbiota. A very interesting example of such scientific works is the manuscript written by Soleimani et al. [14]. The authors of this work indicated that ballast waters are a very important element in the transmission of *Escherichia coli* and *Vibrio cholerae* bacteria. Considering the standard procedures are time-consuming, the authors decided to use molecular methods to search for *E. coli* and *V. cholerae* in their samples. Rapid detection and the identification of pathogens from PCR technique led them to find the presence of *E. coli* and *V. cholerae* in 19 and 14 of the 34 ballast water samples tested, respectively. The results of this study indicated the presence of pathogenic microflora in ballast waters. It also indicated PCR assays as an appropriate alternative to conventional culture methods, which provide quick and accurate data for human health protection procedures [14]. Additionally, the presence of high levels of metabolites and organic matter in the water of ballast tanks create favorable conditions for the development of opportunistic bacteria groups, which leads to a reduction in microbiological competition within these closed spaces [15,16,17,18,19]. New interpopulation interactions may promote disruption of the pattern of interspecies distribution in local waters. These changes may result in the introduction of microorganisms into new regions where, under favorable conditions, they may acquire invasive traits [15,17].

Despite the low number of reports, we should also bear in mind that ballast water can be a source of chemical pollution [19,20,21,22]. High concentrations of heavy metals (e.g., Cr, Pb, and Hg) and ions in ballast water may affect the health of aquatic organisms that are subsequently consumed by humans. Chemical contamination of waters is a global threat as such pollutants can be extremely toxic even at low concentrations [21,23]. Soleimani et al. [20] indicated worryingly high levels of chloride (Cl) and fluoride (F) in ballast water samples coming from ports around the world; the levels of these elements were correlated. In addition, the authors pointed out the necessity of testing seafood obtained from areas around ports for contaminants (ions and heavy metals). They emphasized the significance of the enforcement of international restrictions that standardize the rules of ballast water control and ballast water treatment (WBT) systems created for this purpose. The physical and chemical conditions of water affect the structure, density, and distribution of aquatic microorganisms [24]. Yoshimura et al. [25] found that the salinity, oxygenation, and nutrient content of water have a significant effect on microorganisms. Studies have shown that saline determines the spatial and seasonal distribution patterns of microorganisms in water. Increased saline contributed to a significant decline in total bacterial numbers [26]. Oxygen and nutrients also play an important role in the life of microorganisms. As a result of climate change and increased water pollution, there is a decrease in oxygen and an increase in the content of nutrients. Spietz et al. [27] found a strong negative relationship between bacterial richness and dissolved oxygen. The composition of the bacterial community in all samples was also strongly related to the gradient of dissolved oxygen, and significant changes in the composition of the bacterial community occurred at the concentration of dissolved oxygen between 5.18 and 7.12 mg O2 L-1. This dissolved oxygen threshold value was higher than the classical definitions of hypoxia (<2.0 mg O2 L-1), suggesting that changes in bacterial communities might precede harmful effects on higher vertebrates, such as fish and aquatic mammals [27]. It is widely believed that it is important to test ballast waters in order to protect the sanitary conditions of port waters and the health of aquatic species [9,14,16,17].

The aim of this study was to investigate the impact of seasonal habitat conditions on the structure of the microbiota in the ballast water of long and short-range ships with treatment systems in place, entering the Police Seaport in Oder estuary.

## 2. Materials and Methods

### 2.1. Sampling

Ballast water samples were obtained from eight cargo, general cargo, and bulk carrier ships that were 90 to 200 m long and had docked at the Odra River estuary sea port of Police in the southern Baltic Sea in the Republic of Poland (53°33′44.8″ N 14°35′15.2″ E). Short-range ships (S1–S4) were considered to be those that operated in the area of Europe’s territorial waters, namely, the Baltic, North, and Mediterranean Seas. Long-range ships (L1–L4) had traveled over more extensive sea lanes (the Atlantic Ocean). Regardless of the extent of the ship’s voyage, ballast waters were also divided by the timing of their collection into autumn (S1 and S2 and L1 and L2) and winter (S3 and S4 and L3 and L4). Port waters were also sampled in autumn (P1) and winter (P2). Each sample consisted of several subsamples and was taken three times. Replicates were included in the results, and standard deviations were calculated. It should be mentioned that, in accordance with BWM regulations and local rules, all of these vessels exchanged ballast water before entering the port. In order to ensure the representativeness of the samples, the sampling process was performed in accordance with the IMO International Guidelines for Ballast Water Sampling [21]. Samples were taken approximately 15 min after the ballast tanks were opened while the ship was stationed at the dock, and the process itself involved pumping water through the ballast water pump until suction was lost. The volume of each bulk sample taken was divided equally for each analysis that was planned. In order to obtain a homogeneous mixture, each of the contamination-protected 20 L samples was mixed before distribution. Water samples from the ballast tanks in individual seasons were matched with port water samples taken from the Police Sea Port area (South Baltic estuary), which served as a reference for local waters.

### 2.2. Analysis of Physical and Chemical Properties of Samples

Collected water samples were transported to the laboratory for physicochemical analyses. pH, electrolytic conductivity (EC), and dissolved oxygen (DO ppm) were measured with an HQ30D multiparameter from Hach (Düsseldorf, Germany); salinity–PSU was measured with an HI 96821 digital refractometer (Olsztyn, Poland); salinity classification was measured against the Venice system; electrochemical potential (mV) was measured with a Greisinger GMH 3511 set meter (Regenstauf, Germany); total hardness (oN) was measured by a titration method based on PN-ISO 6059:1999; and concentrations of nitrogen forms and phosphate were measured by a colorimetric method using an automatic flow analyzer from Skalar (Breda, The Netherlands) according to PN-ISO 7150-1:2002, PN-EN ISO 10304-1:2009, and PN-EN ISO 6878:2006.

### 2.3. Microbiological Analysis of Samples 

Microbiological analysis was performed on water samples from the port area (P1 and P2) and ships’ ballast water (S1–S4 and L1–L4), taking into account the seasonal distribution of samples. All microbiological analyses were performed in accordance with accepted standards (ISO 6887-1). Applicable ISO standards were used for the determination of specific microbial groups, namely, ISO 6222:2004P for total bacterial count (TAMC); ISO 7954 for total yeast and mold counts (TYMC and TMMC); and EN-12780 for total *Pseudomonas* counts (TCPs), total halophilic count (THC) on SeaWater medium (12.5% NaCl, 0.0385% KCl, 0.24% MgCl_2_, 0.055% NaHCO_3_, 0.175%), total proteolytic count (TPC: 0.2% yeast extract, 1.0% casein, 0.1% skim milk, 1.5% agar), total lipolytic count (TLC: 0.5% peptone, 0.1% yeast extract, 0.4% NaCl, 1.5% agar, and 0.1% rhodamine B solution and Tween 80 as a carbon source), and total amylolytic count (TAC: 0.1% dextrose, 0.7% K_2_HPO_4_, 0.2% KH_2_ PO_4_, 0.01% MgSO₄, 0.05% Na_3_C_6_H_5_O_7_, 0.1% (NH₄)2SO_4_, 2% starch, 1.5% agar). 

All microbiological tests were carried out by the culture method using media and incubation conditions suitable for a specific group of microorganisms. The number of microorganisms determined was expressed as the log cfu (colony-forming units) value converted per 1 mL of water tested. 

### 2.4. Static Analysis 

The obtained results, both microbiological and chemical, were subjected to statistical analysis using Statistica 13.3 software. Correlation tests between individual parameters were carried out, and principal component analysis (PCA) was performed [28]. Levels of similarity between the microbiotic profiles of individual samples were determined using UPGMA (unweighted pair group method with arithmetic mean) cluster analysis with Pearson distribution. 

## 3. Results

The results of the hydrochemical analyses of ballast water showed significant quantitative variations depending on the areas from which the ships ran and the season in which the samples were taken (Table 1 and Table 2). Analyses of water pH for port waters were neutral. This parameter was 7.58 for the sample from the autumn period (P1) and 7.75 for the sample from the winter period. Increased values (above 7.91) were found for the waters of short-range (S1–S4) and long-range (L1–L4) ships. 

The salinity of ballast water on the short-range ships in both test seasons (autumn and winter) was lower than that of long-range ships. For the autumn trials of short-range ships (S1 and S2), they ranged from 1.22 to 2.11 per mille, while the autumn ballast water of long-range ships (L1 and L2) had a salinity of 3.34–6.18 per mille. Winter ballast water salinities were 1.22 (S3) and 2.11 (S4) per mille for short-range vessels and 5.15 (L3) and 6.18 (L4) per mille for long-range vessels. 

Analyzing the results of ballast water tests, it was observed that an increase in the electrolytic conductivity of water translated into an increase in total hardness. Particularly high hardness values were recorded during the winter season and ranged from 67.48 °N to 90.10 °N for long-range vessels. Relatively low oxygen values were found in all water samples (Table 1 and Table 2). 

This was confirmed by the low content of phosphate and nitrite ions in the ballast water of long-range ships (Table 1 and Table 2).

Analysis of the abundance of the determined microbiota showed that samples taken in autumn had a statistically significant (*p* < 0.1) higher abundance of microorganisms (log 1.97–2.55) than samples taken in winter (log 1.39–2.27). A statistically significant difference was confirmed in autumn between P1 and both L1 and L2 (*p* < 0.05) and a statistically significantly lower number of microorganisms in winter for L4 (*p* < 0.05) than the other analytes (Table 3).

Regardless of the origin of the samples, the dominant group in the microbiotic structure of the tested water was Gram-negative bacilli belonging to the genus *Pseudomonas*, while spore-forming bacteria dominated the TAMC cultures. When evaluating the enzymatic profile, the isolated microorganisms were characterized primarily by amylolytic and lipolytic activity, regardless of the origin of the sample tested. It should be noted that in the total number of microorganisms, bacteria belonging to *Pseudomonas fluorescens* (TCPsf) and halophilic bacteria (THC) were isolated in significantly lower numbers (*p* < 0.1) than other microorganisms. The highest mold abundance (TMMC) was found in autumn samples L2, S2, and L1 and the lowest in winter samples S3 and L3, with none recorded in samples P2, S4, and L4. Representatives of the genera *Aspergillus, Cladosporium, Mucor,* and *Penicillium* were determined in all analyzed samples. The presence of yeast (TYMC) was observed in autumn samples P1, S1, and L1 and in winter sample S3.

Based on the results summarized above, a cluster analysis was performed. The resulting similarity distribution based on the quantitative and qualitative diversity of the isolated microorganisms indicated that the primary criterion was not the origin of the water samples but rather the season of sampling (autumn and winter). The distinguished clusters were characterized by high intragroup similarity, although there was high dissimilarity between them when analyzed in pairs (Figure 1).

Table 4 shows the correlation between the physicochemical parameters of the water samples and the abundance of cultured microorganisms. High correlation values (R > 0.5) were obtained for the relationship of P-PO_4_ and TYMC, TMMC, THC, TAC, and average of total microorganisms (average of total) as well as for the relationship between oxygen levels and TYMC, TCPsf, THC, TLC, and averaged total. In addition, the lowest but statistically significant correlation was found between water hardness and TYMC, TMMC, TCPsf, THC, TLC, and averaged total, while oxygen was correlated with TYMC, TCPsf, THC, TLC, and averaged total. A similar correlation was observed for salinity and TYMC, TCPsf, THC, TLC, and averaged total as well as for EC and TYMC, TMMC, TCPs, THC, TLC, and average total. It is worth noting that with the increase in P-PO_4_ and O_2_, the counts increased in most groups of microorganisms. In the case of other physicochemical parameters, their increase was usually associated with a reduced abundance of each group of microorganisms (Table 4).

Figure 2 graphically illustrates the results of the PCA analysis by the position of the charge vectors relative to the first two principal components. The first axis explained 38.87% of the total variance of the variables, while the second axis explained 15.16%. Among the analyzed physicochemical parameters, those strongest positively correlated with TPC were REDOX and N-NO_2_. A large positive effect of P-PO_4_ on TYMC, TMMC, TAC, TLC, THC, and the average of total was also found, while there was a negative effect of P-PO_4_ on TPC. In addition, the abundance of TPC was positively influenced by EC, pH, and ballast water hardness. The position of the vectors of N-NH_4_ and N-NO_3_ loads indicated that TCPsf increased with an increase in N-NH_4_ concentrations and a decrease in N-NO_3_. The opposite effect of N-NH_4_ and N-NO_3_ concentrations was noted for TAMC.

## 4. Discussion

Ballast water poses a great threat to aquatic ecosystems. Unlike higher organisms, the abundance of microorganisms carried by ballast water is usually very high [29]. The number of microorganisms and their adaptability to new environmental conditions are the primary factors determining their impact on local ecosystems and the safety of hydrobionts [21,23]. The abundance of microorganisms isolated from ballast water in our study (Table 3) is significantly lower than those reported in published reports [30]. The dynamics of variations in microbial counts is complex and should be considered in the context of a consistent interaction of factors that change over the course of a ship’s voyage. Thus, in assessing microbial counts, the presence and function of permanent ballast tank residents and the effects of ballast water exchange (BWE) should be considered [13]. In this case, it is important to determine predictive relationships conditioned by the interactions between microorganisms co-forming biofilms and developing in the ballast tank sediments and the allochthonous microbiota characteristic of transit waters delineated by the voyage route.

The results of our study do not address the analysis of microbiota and phisiochemical parameters’ changes during the voyage, the importance of which was highlighted in Seiden et al. [31]. It must be admitted that this significantly limited the possibility of interpretation and the ability to understand the influence of factors on the dynamics of changes in the abundance of bacteria. As a result, the stage of temporal growth of microorganisms followed by their slow death [32] or periodic changes in the forming microbiota of ballast water was ignored in this study. According to Seiden et al. [31], the environment of a ballast tank is a place of dynamic physical and chemical changes in which biotic and abiotic changes take place. The mutual relations between microorganisms, the release of DOM, and the duration and route of the voyage are the basic factors. These are important in the final determination of the correlation between microbiota and ballast water parameters. This direction of analysis and interpretations is important when considering the ballast water “incubator” hypothesis. The studies by Tomaru et al. [33] and Gerhard and Gunsch [34] clearly indicate that the richness and diversity in ballast tank water are much higher than in natural marine waters and harbors. Such a relationship was established for samples taken in winter, irrespective of the duration of the voyage. In the case of samples taken in autumn, the numbers of labeled bacteria in the port water were comparable to the values in the ballast waters of short-range ships (S1 and S2), whereas they were higher in samples taken from long-range vessels (L1 and L2). Our results indicate that there is limited scope for establishing general patterns of bacterial abundance in the analyzed environments. Under natural conditions, the dynamics of changes in the bacterial population is complex. The structure of the microbial community and its function depend on the abundance and availability of nutrients, physical parameters, and pressure exerted by, for example, microzooplankton [32]. Therefore, the adaptive abilities of microorganisms themselves are of great importance. The most common response of bacteria to unfavorably changing conditions is the creation of persistent forms, the ability to synthesize antibacterial compounds, and adaptation of metabolic pathways to limited nutrients. The scope of changes in the population profile in ballast water also depends on the water source and probably the seasonal cycle of bacterial development [33]. Khandeparker et al. [30] proved that one of the reasons for the varied abundance of bacteria in ballast water is the residual sediment and water, which are not subject to resuspension and exchange during ballast water exchange in tanks. Their biological components are particularly important as they favor the emergence of epibiota, indicating the release of nutrients from dying phytoplankton and zooplankton.

In our study, we did not confirm a correlation between the length of the ship’s route and the abundance of isolated microorganisms. However, we showed consistency in the microbial profiles determined in ballast water from long- and short-range ships for which the clustering/grouping factor was the season in which the water samples were obtained (Figure 1). In principle, it is difficult to unequivocally identify mechanisms for regulating microbial abundance in ballast water. The determined relationship requires an understanding of the mechanism of links between the dynamics of changes that occur in the ballast water environment and microorganisms. However, their interpretation should not be based on a general pattern of behavior. Observations by Drake et al. [32] indicated a decrease in microbial biomass over the course of a voyage, while the findings by Burkholder et al. [35] showed the abundance being maintained at comparable levels and Tomaru et al. [33] found an increase in microbial numbers. An explanation for this may be sought in the correlation with the chemical profile of the ballast water. As salinity increases, the oxidoreductive potential of water usually increases [28]; however, in our study, the differences in salinity were not pronounced and no such correlations were found. However, they are in contrast to the results of Seiden et al. [31], who confirmed a positive correlation of microbial counts with temperature and a negative correlation with dissolved oxygen. In the study by Jang et al. [36], oxygen values in ballast water were much higher. These differences may have resulted from differences in the studies, and the time of transport may affect differences in water oxygenation. It should be acknowledged that the results of our study do not address the analysis of changes during the voyage, which significantly limits the ability to interpret and understand the influence of factors on the dynamics of changes in the abundance of bacteria. As a result, the stage of temporal growth of microorganisms followed by their slow death [30] or periodic changes in the forming microbiota of ballast water were ignored in this study. The studies by Tomaru et al. [33] and Gerhard and Gunsch [34] clearly indicate that the richness and diversity in ballast tank water are much higher than in natural marine waters and harbors. It should be noted that some of the deposited sediment and water are not subject to resuspension and exchange during ballast water exchange in the tanks. This favors the emergence of epibiota, indicating the release of nutrients from dying phytoplankton and zooplankton.

Nutrient abundance is crucial for the development and differentiation of the ballast water microbiota, particularly due to quantitative changes over the course of a voyage. Of particular importance are the sources of C and N, which, when integrated with physicochemical factors, determine the ultimate mechanisms of differentiation of microbiotic community structures [37,38]. This may be reflected in the variability of nutrient elements and pH in ballast water. In their study, Starliper et al. [39] showed that a lower abundance of cultured bacteria was obtained as the pH of the waters increased. We noticed a similar phenomenon in our results. As pH increased, there was a change in the salinity of the water and the level of biogens, such as NO_2_, NO_3_, NH_4_, and PO_4_. In most areas of the Baltic Sea, waters are increasingly acidic, partly due to the dissolution of carbon monoxide from the air in the water. It is predicted that unless CO_2_ emissions decrease, the acidification of water will continue [40]. As these parameters increased, the abundance of microorganisms decreased (Table 1, Table 2 and Table 3). Such changes in ballast water can have a significant impact on the diversity of the microbiota inside the reservoirs, as noted in the work by Seiden et al. [31]. Our results also indicated that variability in environmental conditions associated with different sampling periods affected microbiotic diversity in the ballast waters. The same was observed by Tomaru et al. [33], who argued that a number of environmental factors play a major role and that low oxygen content may be the reason for reduced microbiota abundance. We also noted that as the amount of dissolved oxygen in the samples decreased, the abundance of cultured microbes decreased, which was particularly evident in all of our winter samples (S3, S4, L3, and L4). It should be noted that along with changes in the concentrations of the physicochemical parameters of ballast water, there was a change in the abundance of particular groups of microorganisms in each of the tested samples, for example, the bacteria of the genus *Pseudomonas* sp., which, as facultative anaerobic bacteria, show the ability to grow under aerobic and anaerobic conditions [41]. In our study, we recorded an increase in their abundance with increased oxygenation recorded in an autumn sample of harbor water (P1) and a winter sample from a short-range ship (S4). We also observed that with a significant increase in the electrolytic conductivity of water, and consequently increased total hardness, the total abundance of bacteria of the genus *Pseudomonas* (TCPs) increased. Similar results were obtained by Feisale and Bennett [42], who found that hard water and the salts it contains stimulated the growth of *Pseudomonas aeruginosa* in the samples tested. Conversely, the aforementioned water parameters reduced the growth of yeast (TYMC), fungi (TMMC), *Pseudomonas fluorescens* (TCPsf), and halophilic bacteria (THC). It can be assumed that this was related to an increase in mineral salts. In the study by Forghani et al. [43], slightly acidic electrolyzed water (SAEW) was shown to be an effective disinfectant against various types of microorganisms. As confirmed by studies on heterotrophic bacteria, ballast water tank treatment systems create by-products in the form of e.g., easily degradable substrates, which lead to subsequent growth of heterotrophic microorganisms in ballast water. This phenomenon is an important element for meticulous control as it may pose a threat to preserving biodiversity.. Water disinfection can alter the composition of bacterial communities through selective recolonization in ballast water or receiving water, thereby affecting bacterial-controlled functions that are important to the marine food web [17].

Comparing the results obtained in our work with other scientific reports on ballast water, we noticed a consistency in the extensive microbial record [17,44]. However, the problem with such research is that microbes are considered ubiquitous. Although some of them do show specific biogeographic preferences, it is difficult to consider them unequivocally non-native in a given ecosystem. Moreover, when colonizing ballast water, they do not always adapt to their new environment or interact with species within autochthonous trophic chains and even less often acquire invasive traits [15,17,45,46]. On the other hand, in the study by Takahashi et al. [44], numerous aggregations of diverse microorganisms in ballast water were found, including the indicator microorganisms coliforms, enterococci, *E.coli*, and pathogenic strains of *Clostridium* perfringens and *Vibrio cholerae*. Similarly, Ng et al. [47] showed the abundant presence of microorganisms from groups such as *Proteobacteria*, *Cyanobacteria*, *Bacteroidetes*, *Planctomyces*, *Chloroflexi*, *Actinobacteria*, and *Aquificae* as well as pathogens such as *Vibrio spp*. and *Salmonella spp.* in ballast water. All this indicates the significant potential threat associated with ballast water, which is even more serious considering the observed resistance of potentially pathogenic microorganisms to current ballast water treatment methods. Given that newly introduced microorganisms (including opportunistic groups) can compete for resources to varying degrees in new environments [48], they can disrupt local interactions between aquatic organisms and their symbiotic microorganisms, which can lead to a drastic reduction in the resilience of native species and loss of typical biodiversity in these ecosystems [49,50,51,52].

## 5. Study limitations

This study has some limitations. The biggest limitation in our study was the process of collecting samples. As unauthorized persons with lack of appropriate training, we were unable to take the samples ourselves. In addition, the COVID-19 pandemic and the related safety restrictions against the spread of the virus prevented us from direct sampling and supervision of the process (we were not allowed to board the ship). We were reliant on the help of trained sailors. Moreover, the research design could be more specific. Unfortunately, despite ambitious plans to thoroughly analyze the obtained ballast water samples, our research design depended on the equipment available to us. Therefore, we are aware of many shortcomings, which we plan to correct in future research. Another limitation of our work was the team’s inexperience in working with ballast water. This work is only the beginning for us in this field of research. When developing our action plan, we made use of the available literature. We now know that we should streamline and specify each stage of the work more precisely, starting with creating a very detailed methodology. We hope that with the experience gained in this study, improved methodology, and wider access to equipment, we will be able to conduct richer and better organized research.

## 6. Conclusions

Our results showed that all the examined ballast water samples were positive for the presence of representatives of the genera *Aspergillus*, *Cladosporium*, *Mucor*, *Pseudomonas*, and *Penicillium*. However, heterotrophic bacteria and mold fungi dominated in autumn (log 2.45–3.26), while the highest number of *Pseudomonas* bacteria was recorded in winter (log 3.32–4.40). Moreover, the ballast water collected in the first period was characterized by a statistically significantly higher number of microorganisms (log 1.97–2.55) than in the winter period (log 1.39–2.27).

The number of individual groups of microorganisms was determined by the physiochemical properties of samples. The changing concentrations of physicochemical factors significantly influenced the concentration of individual representatives of microorganisms in the tested samples. The increase in the count of most microorganisms was positively corelated with REDOX, EC, O_2_, P-PO_4_, and N-NO_2_. On the other hand, reduced abundance of each group of microorganisms was correlated with water hardness, salinity, and pH. It is worth noting that the position of the vectors of N-NH_4_ and N-NO_3_ loads indicated that TCPsf increased with an increase in N-NH_4_ concentration and a decrease in N-NO_3_ concentration. The opposite effect of N-NH_4_ and N-NO_3_ concentrations was noted for TAMC. In the case of other physicochemical parameters, their increase was usually associated with reduced abundance of each group of microorganisms. This study can be useful as a database for further, more detailed studies on the biological profile of ballast water and deduction of its impact after discharge in the South Baltic area. In addition, further studies based on the search for ships from very cold or very saline regions can serve as very important elements in examining the process of preserving microorganisms from untypical areas for settlement in waters and ecosystems connected to the Baltic Sea. Therefore, it seems important to pay attention to a broader analysis of the microbiological status of both ballast water and Baltic Sea water. Such a research goal will allow more precise determination of the effectiveness of ballast water and sediment treatment systems installed on ships, especially in terms of eliminating undesirable microorganisms, particularly those with pathogenic potential. We would like to emphasize the need for further research to evaluate the risk of microorganisms from ballast water in new environments in terms of their ability to establish interactions with higher trophic levels and how they affect native species of aquatic organisms, especially those of high economic importance.

## Figures and Tables

**Figure 1 ijerph-19-15598-f001:**
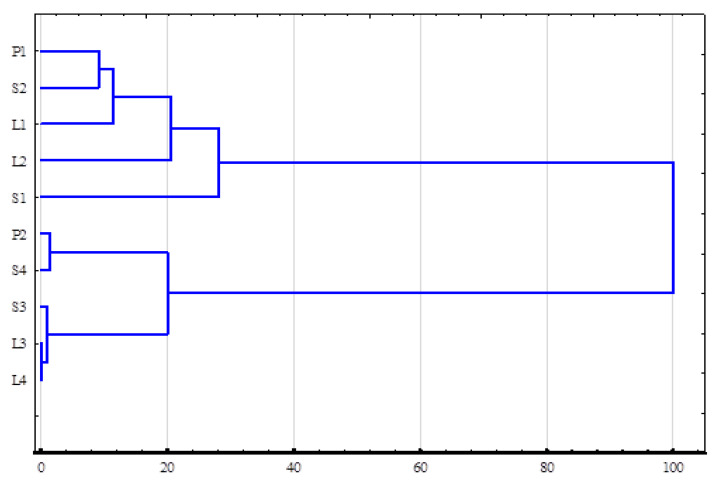
Dendrogram of cluster analysis of the similarity of the microbiological profile of analyzed samples; P—Police Sea port waters: P1, autumn and P2, winter; S—short-range vessel: S1–S2, autumn and S3–S4. winter; L—long-range vessel: L1–L2, autumn and L3–L4, winter.

**Figure 2 ijerph-19-15598-f002:**
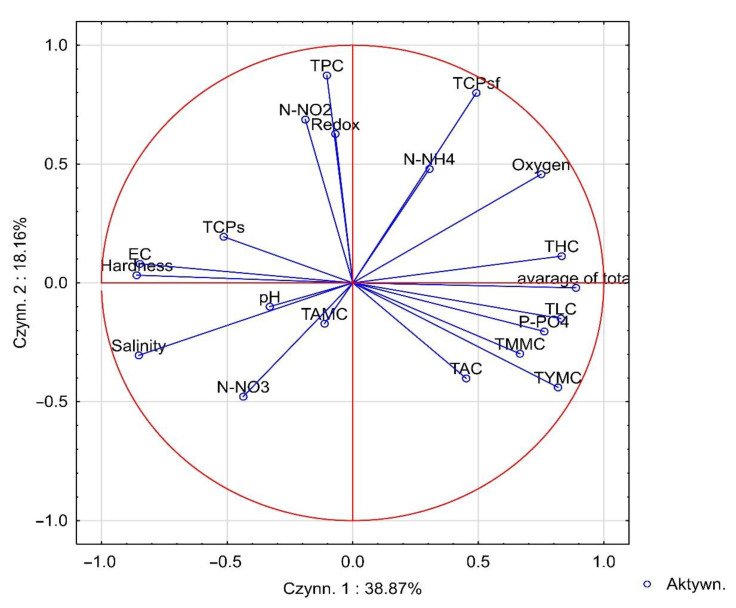
Graphical interpretation of principal component analysis.

**Table 1 ijerph-19-15598-t001:** Physical and chemical properties of autumn control and ship ballast water samples.

	Sample		P1 (SD)	S1 (SD)	S2 (SD)	L1 (SD)	L2 (SD)
Property							
pH			7.58 (0.57)	8.03 (0.18)	8.00 (0.02)	8.10 (0.02)	8.02 (0.20)
PSU (‰)			0.34 (0.23)	1.35 (0.23)	1.43 (0.96)	3.34 (1.19)	4.94 (2.03)
Venice system for the classification of water salinity			Limnetic	Mixohaline	Mixohaline	Mixohaline	Mixohaline
Oxidoreductive potential (mV)			185 (0.01)	181 (0.02)	177 (0.01)	176 (0.03)	169 (0.02)
Electrolytic conductivity EC (mS/cm)			9.31 (0.58)	4.92 (0.08)	5.36 (0.18)	6.38 (0.49)	6.38 (0.21)
Total hardness (°N)			46.88 (0.12)	27.51 (0.15)	29.83 (0.03)	32.48 (0.05)	36.14 (0.11)
Dissolved oxygen (mg L^−1^)			4.60 (0.16)	2.80 (1.04)	3.10 (0.76)	1.60 (0.06)	1.80 (0.07)
Levels of biogenic elements [mg/L]		NO_2_	0.04 (0.00)	0.02 (0.01)	0.04 (0.00)	0.04 (0.00)	0.05 (0.01)
		NO_3_	2.26 (0.22)	4.30 (0.19)	2.77 (0.47)	1.91 (0.38)	1.23 (0.03)
		NH_4_	0.31 (0.03)	0.28 (0.06)	0.26 (0.03)	0.34 (0.06)	0.29 (0.01)
		PO_4_	1.03 (0.51)	2.02 (0.40)	1.63 (0.23)	0.77 (0.18)	0.65 (0.29)

**Table 2 ijerph-19-15598-t002:** Physical and chemical properties of winter control and ship ballast water samples.

	Sample		P2 (SD)	S3 (SD)	S4 (SD)	L3 (SD)	L4 (SD)
Property							
pH			7.75 (0.04)	7.91 (0.08)	8.08 (0.04)	8.07 (0.12)	7.93 (0.17)
PSU (‰)			0.47 (0.62)	1.22 (0.46)	2.11 (1.30)	5.15 (2.16)	6.18 (2.34)
Venice system for the classification of water salinity			Limnetic	Mixohaline	Mixohaline	Mixohaline	Mixohaline
Oxidoreductive potential (mV)			196 (0.02)	189 (0.00)	195 (0.01)	183 (0.00)	185 (0.00)
Electrolytic conductivity EC (mS/cm)			7.65 (1.56)	2.00 (0.30)	9.31 (1.90)	14.99 (0.47)	16.37 (2.09)
Total hardness (°N)			30.65 (0.01)	19.01 (0.02)	57.22 (0.05)	67.48 (0.02)	90.10 (0.04)
Dissolved oxygen (mg L^−1^)			4.30 (0.05)	2.10 (0.56)	3.00 (0.09)	1.10 (0.56)	0.60 (0.06)
Levels of biogenic elements [mg/L]		NO_2_	0.05 (0.01)	0.04 (0.01)	0.34 (0.42)	0.06 (0.01)	0.06 (0.01)
		NO_3_	2.08 (0.88)	4.67 (2.51)	2.52 (1.83)	5.11 (0.74)	4.82 (0.60)
		NH_4_	0.31 (0.02)	0.21 (0.01)	0.30 (0.05)	0.34 (0.09)	0.13 (0.01)
		PO_4_	0.63 (0.26)	0.34 (0.31)	0.58 (0.36)	0.15 (0.02)	0.08 (0.05)

**Table 3 ijerph-19-15598-t003:** Quantitative diversity of microorganisms isolated from water samples.

Sampling Collection		Microbiological Fractions (log_10_/mL)									
		TAMC	TYMC	TMMC	TCPs	TCPsf	THC	TLC	TAC	TPC	Average of Total
Autumn	P 1	4.61	2.98	2.45	4.11	1.30	1.15	2.90	2.78	0.70	2.55 ± (0.44) ^a^
	S 1	4.45	2.84	2.78	3.08	1.18	1.40	2.18	3.53	1.00	2.49 ± (0.38)
	L 1	3.98	2.90	3.26	3.49	0.90	0.85	2.30	3.02	0.90	2.40 ± (0.41) ^b^
	S 2	4.19	0.00	3.09	3.41	1.70	0.00	1.93	1.88	1.88	2.01 ± (0.40)
	L 2	3.70	1.00	3.19	3.08	1.08	0.70	2.48	2.00	0.48	1.97 ± (0.40) ^ab^
Winter	P 2	3.29	1.00	0.00	3.32	2.00	1.40	1.70	2.15	2.10	1.77 ± (0.47) ^c^
	S 3	4.19	2.71	1.48	4.03	1.00	1.08	2.93	2.81	0.00	2.25 ± (0.48) ^d^
	L 3	4.57	0.00	1.30	4.40	2.65	0.70	1.85	2.57	2.39	2.27 ± (0.51) ^c^
	S 4	4.18	0.00	0.00	4.18	0.00	0.00	1.48	2.36	1.18	1.49 ± (0.58) ^d^
	L 4	4.42	0.00	0.00	4.23	0.00	0.00	0.70	2.49	0.70	1.39 ± (0.61) ^cd^

^a,b,c,d^—TAMC—total bacterial count; TYMC—total yeast count; TMMC—total mold count; TCPs—total *Pseudomonas* count; TCPsf—total *Pseudomonas fluorescens* count; THC—total halophile count; TLC—total lipolytic bacterial count; TAC—total amylolytic bacterial count; TPC—total proteolytic bacterial count.

**Table 4 ijerph-19-15598-t004:** Correlation matrix between physicochemical parameters of the ballast water and the microbiota.

Variable	TAMC	TYMC	TMMC	TCPs	TCPsf	THC	TLC	TAC	TPC	Average of total
pH	0.07	**−0.46**	0.16	−0.13	−0.06	**−0.52**	−0.31	−0.1	0.21	−0.2
Oxygen	−0.14	**0.55**	0.18	−0.15	**0.61**	**0.8**	**0.51**	0.27	0.25	**0.64**
EC	0.23	**−0.63**	**−0.65**	**0.55**	**−0.46**	**−0.63**	**−0.78**	−0.26	0.17	**−0.72**
Salinity	0.05	**−0.69**	−0.26	0.16	**−0.6**	**−0.84**	**−0.63**	**−0.45**	−0.15	**−0.76**
Hardness	**0.38**	**−0.61**	**−0.59**	**0.64**	**−0.43**	**−0.65**	**−0.76**	−0.18	0.09	**−0.64**
Redox	0.06	−0.1	**−0.69**	**0.53**	**0.4**	0.31	−0.19	0.12	**0.43**	−0.06
N−NO_2_	**0.32**	**−0.43**	−0.2	**0.52**	**0.58**	−0.09	−0.17	−0.07	**0.58**	0.08
N−NO_3_	**0.39**	0.02	**−0.52**	**0.44**	**−0.66**	−0.21	**−0.34**	**0.47**	−0.36	0.31
N−NH_4_	−0.18	−0.1	0.25	−0.26	**0.45**	0.08	0.28	−0.28	**0.52**	0.22
P−PO_4_	0.10	**0.68**	**0.69**	**−0.59**	0.23	**0.57**	**0.41**	**0.66**	0	**0.75**

Statistically significant correlations are indicated in bold (*p* < 0.05).

## Data Availability

Not applicable.

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
