# Peer review of "Habitat Conditions of the Microbiota in Ballast Water of Ships Entering the Oder Estuary"

_ijerph, 2022, doi:10.3390/ijerph192315598_

Round 1
Reviewer 1 Report
Manuscript entitled “HABITAT CONDITIONS OF THE MICROBIOTA IN BALLAST WATER OF SHIPS ENTERING THE ODER ESTUARY” submitted by Kinga ZatoÅ„-Sieczka et al. can be accepted for publishing in International Journal of Environmental Research and Public Health after major revisions.
Comments to the Author:
1. Please avoid using keywords that already exist in the title.
2. The authors have mentioned in the lines 255 and 256 that "we did not confirm a correlation between the length of the ship’s route and the abundance of isolated microorganisms". They can make this claim if they have checked the abundance of microorganisms in the source port. It is not possible to judge about the decrease and/or increase the abundance of microorganisms in ballast water along the ship's route if the source port water data is not available. Please revise this section.
3. In the discussion, the authors should compare the quality of the entered ballast water with the sea water of the destination (Police sea port waters) to determine whether the discharge of these waters is a threat to the destination port and the ecosystem of this area or not.
4. In the conclusion part, the author should simply and clearly mention the main results. Please improve the conclusion section.
5. In the introduction where applicable refer to the following related studies on ballast water:
Molecular detection of E. coli and Vibrio cholerae in ballast water of commercial ships: a primary study along the Persian Gulf. Journal of Environmental Health Science and Engineering, 19(1), pp.457-463.
Heavy metal levels of ballast waters in commercial ships entering Bushehr port along the Persian Gulf. Marine Pollution Bulletin, 126, 74-76.
Assessment of microbial and physiochemical quality of ballast water in commercial ships entering Bushehr port, along the Persian Gulf. Desalination and Water Treatment. 98, 190-5.
Assessment of microbial and physiochemical quality of ballast water in commercial ships entering Bushehr port, along the Persian Gulf. Desalination and Water Treatment. 98, 190-5.
- Please add some recommendations for future works.
- Please add a section of study limitation before conclusion.
Author Response
All author's coverletter added as pdf data

Reviewer 2 Report
i) Ensure that all the genus and species names are written in italics.
ii) The following sections of the results need to be revised because they have an aspect of discussion:- Lines 141-144; Lines 151-153; Lines 159-161 and Lines 163-164.
iii) In lines 270-271, there should be an explanation of why the results differ from those reported by Seiden et al.

Author Response

(The authors gave the same response as above.)

Reviewer 3 Report
The manuscript by ZatoÅ„-Sieczka et al., is a description of the physicochemical and microbiological properties of ballast water from long- and short-range ships entering a southern Baltic port. The major flaw of the manuscript is the experimental design with a limited number of samples and no clear aim of the study. The introduction fails to present a specific aim for the study and there is no clear conclusion. Although there is a specific section for the conclusion of the study the authors do not use this section to present a conclusion, they use it only to “emphasize the need for further research on the importance of microorganisms in new environments”. Therefore, it is hard to glean from the manuscript what its contribution was to the field. The manuscript lacks innovation given that there are multiple studies on the same subject.
The limited number of samples considered, only 2 negative controls (port water) and only 8 test samples, which are further divided in 2 by season or by travel range result in only marginal significance of the results (p<0.1) which hinder the authors’ ability to give a clear conclusion of the study.
There are some questions about the results as well. For example, the dendrogram, Fig A1., has 5 branches for the autumn season as expected but only four for the winter cluster. Which winter sample was omitted and why? The dendrogram needs to be redone!
The results for the dissolved oxygen are also questionable, why is the concentration of dissolved oxygen lower in wintertime compared to autumn? Cold water can hold more dissolved oxygen than warm water. How do the authors explain the lower dissolved oxygen concentration in winter, when water is colder compared to autumn when water temperatures are warmer?
Although port water was sampled there are no comparisons between the samples taken from the port and the ballast samples.
The article cannot be published in the current format. I recommend a major revision that includes an increased number of samples (at least a multiple of 3 for each condition) that will hopefully aid in generating some clear conclusions.
Author Response

(The authors gave the same response as above.)

Round 2
Reviewer 3 Report
1. The authors indeed strived to write an aim of the study but it is too ambiguous and does not cover the complete aim of the study. I believe a more accurate aim is: to investigate the impact of seasonal habitat conditions on the structure of the microbiota in the ballast water.
If the aim was to look only at habitat conditions why did the authors bother to count bacteria and determine their properties?
The conclusion is still too general and needs to be more specific. The only conclusion given by the authors is “The changing concentrations of physicochemical factors significantly influenced concentration of the individual representatives of microorganisms in tested samples.” which I find too general. Authors should specify which physicochemical factors influenced what microorganisms.
3. I can see the dendrogram in the response to reviewers but it is now completely missing from the revised manuscript, make sure to include it in the manuscript as well.
Some comments on the newly introduced text:
I do not understand why the paragraph, in the introduction, about heavy metals has been expanded. This paragraph was not necessary in the first place since authors do not address this issue in their study. The emphasis on these contaminants only points to the limitation of this study which did not measure such contaminants. I believe more importance should be given to other habitat condition like salinity, oxygen, and nutrients (habitat conditions that were actually measured in the study) and how they are similar or different in ballast water and port water and how these habitat conditions could select for certain microorganisms that would then contaminate the port water. Such conditions have been studied in the manuscript but are not address in the introduction. However they are amply addressed in the discussion therefore, I suggest to move part of it from the discussion to the introduction. In this way the paper will be more cohesive.
Lines 82-83 the phrase “even at low concentration” is repeated twice in the same sentence, one should be deleted.
Author Response
data in attachment
